# Uncertainty Estimation Using a Single Deep Deterministic Neural Network - ML Reproducibility Challenge 2020

## Reproducibility Summary

**Scope of Reproducibility**

The investigated paper claims RBF network when trained with BCE loss along with two-sided gradient penalty outperforms deep ensemble in the task of out of distribution(OoD) detection along with competitive accuracy to softmax based models. The Paper claims to outperform AUROC on Fashion-MNIST vs MNIST and CIFAR-10 vs SVHN. The proposed algorithm is reported to detect both aleatoric and epistemic uncertainty as OoD. Authors mention the need for a formal way to distinguish between the two kinds of uncertainty and pose it as an interesting future research avenue.

The scope of this report is to reproduce the results related to OoD detection presented in the paper. Along with the reproduction of results, we propose an extension for explicit detection of aleatoric and epistemic uncertainty as intended by the authors.

**Methodology**

The author's training code is available on GitHub. Additionally, we have made available all the experimentation codes in the form of notebooks. We provide all the results and analysis on the models described in the paper and trained on NVIDIA Tesla T4 GPU.

**Results**

Overall our reproduction supports the claims of the paper, we can replicate trends and plots as described in the paper. Most of the results are within 1% of the value reported. Notably, AUROC(M) of DUQ in OoD detection of Fashion-MNIST vs MNIST is off by 1.5% and we got a different optimal value for gradient penalty weight ($\lambda$) in it. Also, the results of our proposed extension and its analysis are encouraging. Our proposed extension provided an increase of 1.8% in AUROC(M) in Fashion-MNIST vs MNIST and provided explicit control over the aleatoric and epistemic uncertainty.

**What was easy**

The proposed approach is quite simple and elegant. The availability of the author's code made the implementation of various experiments easy.

**What was difficult**

Understanding of proposed approach requires advanced knowledge of calculus related to the Lipschitz constant and its role to quantify the upper bound of the sensitivity of any function.

**Communication with original authors**

We discussed our report with the authors, they find our analysis on aleatoric uncertainty interesting and appreciate our proposed extension and its encouraging results.

## 1   Introduction

Reliable uncertainty quantification has been a challenge in deep learning, models giving overconfident wrong predictions can not be deployed for practical purposes and if deployed can even be fatal. The investigated paper Uncertainty Estimation Using a Single Deep Deterministic Neural Network [1] presents a method of training deep neural networks to detect out of distribution(OoD) points in a single forward pass along with classification.

Authors define a set of feature vectors called centroids each corresponding to a certain class. Distance between feature vectors predicted by model and centroids are used for class prediction. This architecture along with BCE loss employs a two-sided-gradient penalty to make the model sensitive to input and hence providing OoD detection ability, this approach is called deterministic uncertainty quantification, DUQ.

## 2   Scope of reproducibility

The proposed algorithm outperforms the state-of-the-art techniques for OoD detection on Fashion-MNIST vs MNIST, CIFAR-10 vs SVHN in terms of AUROC as well as computational time. DUQ, an RBF network-based algorithm provides competitive accuracy to softmax models along with strong OoD detection ability. The scope of this report is as follows.

### 2.1   Addressed claims from the original paper

- OoD detection ability on various datasets as reported in paper.
- Role of different hyperparameters (Ablation Study).

### 2.2   Other experiments

- Noise sensitivity of proposed algorithm.
- Propose extension E-DUQ for explicit detection of aleatoric and epistemic uncertainty .

## 3   Methodology

### 3.1   Model descriptions

Figure 1 represents the complete algorithm, first, the features are computed through a standard feature extractor ($f_\theta$) after this extracted features are passed through class-specific layers ($W_c$) to calculate the feature vector for each class. Distance ($K_c$) of this vector from the centroid in a kernel space represents uncertainty in prediction. The centroid is calculated for each batch as described in the equations below.

$n_{c,t} = \gamma \times n_{c,t-1} + (1-\gamma) \times n_{c,t}$

$m_{c,t} = \gamma \times m_{c,t-1} + (1-\gamma) \times \sum W_c f_\theta(x_{c,t,i})$

$e_{c,t} = m_{c,t} \, / \, n_{c,t}$

Where $n_{c,t}$ is number of images of class $c$ in a particular batch at time $t$, $m_{c,t}$ is weighted sum of feature vectors through different mini batches and $\gamma$ is the momentum for updating centroids. $x_{c,t,i}$ is the element i of minibatch at time t of class c.

Total cost comprises of two-loss functions, one is the BCE loss which is the sum of cross-entropy of a binary one-hot vector with the actual label of that class as described in the equation below.

$L_1(x,y) = \sum_c y_c \log(K_c) + (1 - y_c) \log(1 - K_c)$

Usually, deep learning models suffer from $feature\ collapse$ where the presence of some non-robust discriminative features in input space will map inputs to same outputs, the degree to which a model prevents this feature collapse is its

74 *sensitivity*, a two-sided gradient penalty loss is used to enforce sensitivity to our model which is defined as:

75

76 Two-sided GP loss = $\lambda \cdot \left[ \|\nabla_x \sum_c \mathrm{K}_c \|_2^2 - 1 \right]^2$

77

78 Which essentially keeps the norm of Jacobian above a threshold and prevents feature extractor from collaps-
79 ing to a constant function.

80 Another version of gradient penalty can be a single-sided loss as:

81

82 Single-sided GP loss = $\lambda \cdot max(0, \|\nabla_x \sum_c \mathrm{K}_c \|_F^2 - 1)$

83

84 The investigated paper claims two-sided gradient penalty can enforce sensitivity in the model while a single-sided
85 penalty is not able to do so. We will verify these claims empirically in sections to go.

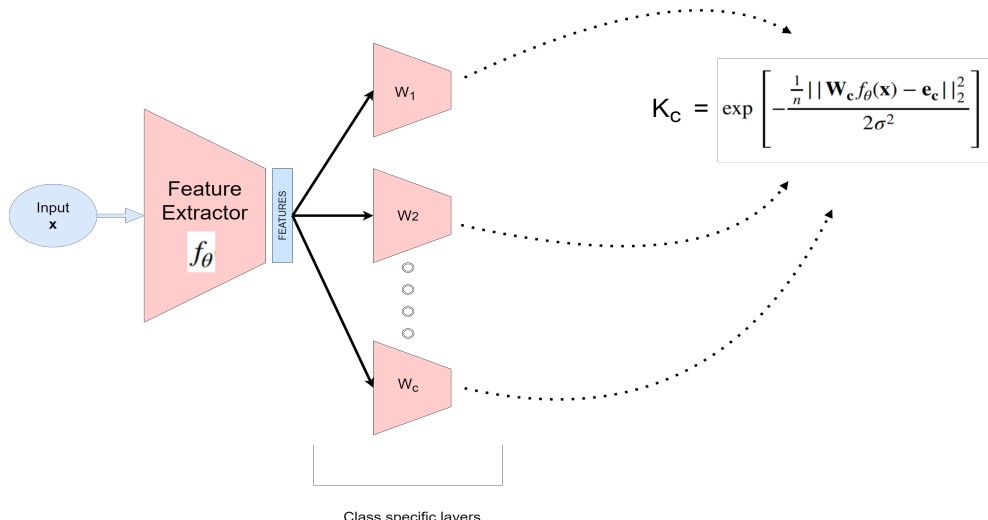

Figure 1: Figure describing complete algorithm, for each class it computes RBF score ($K_c$) which is the measure of OoD detection

86 ## 3.2 Datasets

87 ### 3.2.1 Two-moons dataset

88 It is a two-dimensional dataset. As the name suggests, it has two intertwined classes on the shape of the crescent. We
89 obtain it using Sciket-learn implementation as described by the authors.

90 ### 3.2.2 Fashion-MNIST and MNIST

91 Fashion-MNIST is a dataset of 28x28 grayscale images consisting of a training set of 60,000 images and a test set of
92 10,000 images each belonging to either of 10 classes. MNIST is a database of 28x28 grayscale handwritten digit images
93 having a training set of 60,000 and a test set of 10,000 images. We obtain both datasets using Pytorch's dataset module.
94 As we are testing the proposed algorithm on out-of-distribution detection Fashion-MNIST, MNIST has been a notably
95 difficult pair for this task [2].

96 ### 3.2.3 CIFAR-10 and SVHN

97 CIFAR-10 is a dataset of 60,000 32x32 colour images in 10 classes, having 50,000 training images and 10,000 test
98 images. SVHN is a real-world image dataset similar in flavor to MNIST but incorporates an order of magnitude more
99 labeled data with 604,388 training set images and 26,032 images for testing.
100 Similarly we will evaluate algorithm on CIFAR-10 vs SVHN OoD detection [2].

### 3.3 Experimental setup

The author's training code is available on GitHub which was quite useful for us. We have added our codes for all analysis and experimentation on `https://drive.google.com/drive/folders/1bBrn2jRnRTIhxrAi7BXLOPkzCuUKlizF?usp=sharing`. We trained all models on NVIDIA Tesla T4 GPU.

Execution of the code is not computationally extensive. In the described experimental settings training on MNIST takes 15 minutes while training on CIFAR-10 takes 4 hours. Maximum 3 GB of GPU and 4 GB of RAM is sufficient to run the code.

We followed the Appendix of the reported paper for architecture, optimization, the number of epochs, batch size and generating data in the case of two-moons.

Additionally, we also find and recommend applying early stopping in training on CIFAR10 at around 60 epochs to avoid over-training.

## 4 Results

In this section, we provide our empirical analysis on claims in Section 2, first on toy dataset, two-moons, and then extend the analysis for Fashion-MNIST and CIFAR-10. Our experiments verify the claims of paper with similar results on these datasets.

### 4.1 Two-moon

Our implementation on two-moons for the deep ensemble, DUQ without gradient penalty, with single and with a two-sided penalty are shown in Figs 2-5. The yellow colour represents the region of certainty while the blue region shows uncertainty.

It can be seen DUQ with a two-sided penalty is certain only near the training data and uncertain elsewhere, hence showing the best uncertainty estimation over the region as compared to other methods. These figures validate the importance of the two-sided penalty in enforcing sensitivity.

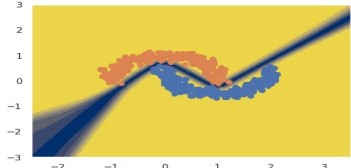

Figure 2: Uncertainty estimation using Deep Ensemble

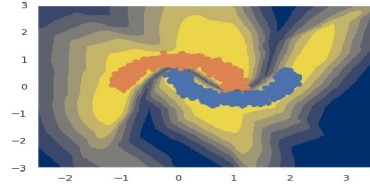

Figure 3: Uncertainty estimation using DUQ with $\lambda=0$

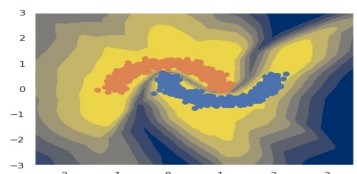

Figure 4: Uncertainty estimation using DUQ with single sided gradient penalty

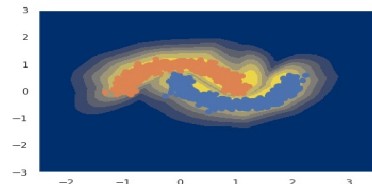

Figure 5: Uncertainty estimation using DUQ with two-sided penalty

### 4.2 Fashion-MNIST

DUQ is trained on Fashion-MNIST and its ability to detect MNIST datapoints as OoD is measured by AUROC, larger being the better. In this section, we provide our results and analysis for DUQ on Fashion MNIST claimed by the authors.

| Method | Accuracy(FM) | AUROC(M) | Train time(in sec) | Inference time |
|---|---|---|---|---|
| Deep Ensemble | 93.30 ±0.32 | 0.889 ±0.005 | 45 | 2.3 |
| DUQ ($\lambda = 0.05$) | 92.13 ±0.29 | 0.947 ±0.005 | 23 | 1 |
| E-DUQ | 92.35 ±0.15 | 0.964 ±0.005 | 23 | 1 |

Table 1: Result comparison of models trained on Fashion MNIST by different methods(mean over 3 runs), AUROC(M) is for separating Fashion-MNIST from MNIST. EDUQ is described in section 4.2.2.

In Table 1, we compare DUQ with deep ensemble, DUQ outperforms deep ensemble in AUROC as well as on training and inference time. We observe a difference of 1.5% in AUROC of DUQ from the original paper.

In an algorithm that can detect OoD points, another measure of performance could be rejection classification, in which we mix two-datasets and reject points based on uncertainty predicted by the model and expect an increase in accuracy. In Fig 8 we find that rejection classification performance on Fashion-MNIST vs MNIST for DUQ in comparison with deep ensemble gives more accuracy increase per rejection as the area below DUQ curve is more that area below DE. These experiments broadly support the claims made by the authors.

### 4.2.1 Aleatoric sensitivity

Authors provide OoD detection analysis on Fashion-MNIST vs MNIST datapoints which are fundamentally different datasets. The paper mentions that DUQ captures both aleatoric and epistemic uncertainty but doesn't provide any analysis on these datasets.

To evaluate the noise detection sensitivity of DUQ, we plot the rejection classification performance of DUQ on Fashion-MNIST vs Noisy Fashion-MNIST as shown in Fig 6. Noisy Fashion-MNIST is Fashion MNIST added with a zero-centered gaussian noise with an std of 0.05 which is imperceptible to the human eye. We have used full test-dataset (1:1 proportion) for the mix. It can be seen that DUQ can even detect Noisy Fashion-MNIST datapoints with this low noise whereas deep ensemble is unable to do so. This clearly states that DUQ is very sensitive to noise and deems even very low noise data points as OoD which can even be undesirable for practical purposes, as extreme sensitivity will harm the model's decision-making ability. This observation is due to the imposed sensitivity to the model by gradient penalty loss. We address this issue in the next subsection and propose an extension.

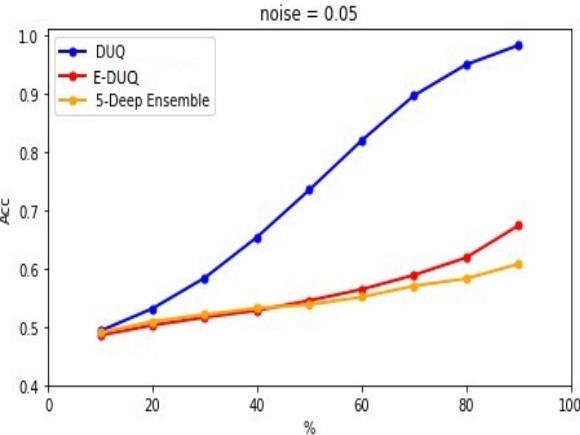

Figure 6: Rejection classification performance by different methods for Fashion-MNIST vs Noisy Fashion-MNIST. (X-axis is the % of data rejected using uncertainty estimates, Y-axis is the corresponding accuracy)

### 4.2.2 Learnable $\sigma$ : Modelling noise through input

We propose to incorporate noise computation by predicting length scale($\sigma$) for each data point rather than keeping it constant as done by [3]. Authors of [3] showed how learning length scale can make model learn to attenuate loss from

erroneous labels. We name this extended approach Explicit-DUQ or E-DUQ as it gives us explicit control over aleatoric and epistemic uncertainty.

If noisy points are predicted with high $\sigma$, it will increase RBF score as shown in Fig 1 hence, disabling the model to predict noisy points as OoD. Hence E-DUQ should label only fundamentally different data distributions (i.e. epistemic uncertainty) as OoD. In E-DUQ, the feature extractor outputs features along with one extra logit for length scale, it will be very similar to the model described in Fig 1, but now along with $f_\theta(x)$ length scale$(\sigma)$ will also be provided by the model to compute $K_c$.

Fig 7 shows the histogram of the magnitude of $\sigma$ predicted by E-DUQ on Fashion-MNIST and Noisy Fashion-MNIST test set, it can be seen that model predicts higher values of $\sigma$ for noisy data points.
Also, Rejection classification plot of E-DUQ (Aleatoric section, Fig 6) lies well below than that of DUQ which validates E-DUQ does not label noisy points as OoD and is robust to noise.

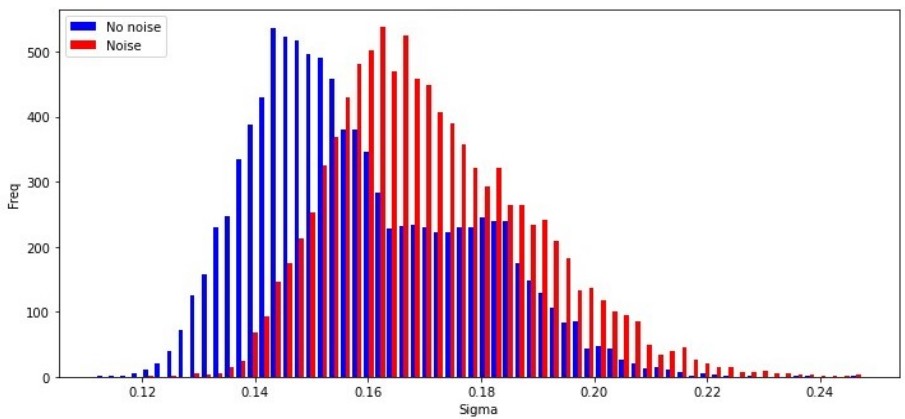

Figure 7: Histogram of values of length scale(sigma) predicted by E-DUQ for Fashion-MNIST and Noisy Fashion-MNIST

Table 1 illustrates E-DUQ outperforms DUQ trained on Fashion-MNIST in AUROC(M), this is because MNIST datapoints are OoD with respect to Fashion-MNIST in terms of epistemic uncertainty, which E-DUQ models explicitly. With the above empirical analysis, we can say it is possible to predict epistemic and aleatoric uncertainties independently using RBF score$(K_c)$ and length scale$(\sigma)$ respectively.

### 4.2.3   Other Hyperparameters and Ablations

In the following tables, ablations are done over centroid dimension, gradient penalty factor, and gradient penalty constant respectively (averaged over 5 runs). All other hyper-parameters are same as given in the original paper.

Table 2 shows how the performance of the model varies with the dimension of the centroid vector ($z$), we note AUROC first increases with $z$ and then saturates while accuracy remains unaffected. Authors used $z = 256$ which according to us is a descent choice.

In Table 3, we show how the performance of DUQ varies with the value of $\lambda$, we observe first increase and then decrease in AUROC score which is also claimed by authors. We find a 1.5% deviation in the best AUROC(M) score as reported by the authors. They also mentioned best AUROC(NM) coincides with best AUROC(M) but we are not able to see this in our reproduction.

We also experiment how varying the constant number$(\alpha)$ in two-sided gradient penalty $\left[\|\Delta_x \sum_c K_c\|_2^2 - \alpha\right]^2$ affect training in Table 4. We fix $\lambda=1$ and experiment for different values for $\alpha$, we find its performance to be better than model trained with optimal $\lambda$ (in table 3). Thus, we infer that $\alpha$ can be a crucial hyperparameter.

Overall ablation study supports the claims of the paper with minor deviations, also we show some important hyperparameters which are not stressed by the authors.

| Size | Accuracy | AUROC(M) |
|---|---|---|
| 10 | 92.15 ±0.15 | 0.887 ±0.015 |
| 256 | 92.13 ±0.29 | 0.947 ±0.005 |
| **500** | 92.32 ±0.15 | 0.959 ±0.004 |
| 1000 | 92.11 ±0.32 | 0.955 ±0.002 |

Table 2: Accuracy and AUROC(M) trend with centroid-vector dimension on Fashion MNIST.

| λ | Accuracy | AUROC(NM) | AUROC(M) |
|---|---|---|---|
| 0 | 92.39 ±0.08 | 0.941 ±0.005 | 0.941 ±0.011 |
| **0.05** | 92.13 ±0.29 | 0.962 ±0.007 | 0.947 ±0.005 |
| 0.1 | 92.11 ±0.08 | 0.953 ±0.006 | 0.942 ±0.007 |
| 0.2 | 92.03 ±0.11 | 0.929 ±0.006 | 0.954 ±0.005 |
| 0.3 | 92.11 ±0.22 | 0.941 ±0.016 | 0.946 ±0.008 |
| 0.5 | 91.94 ±0.16 | 0.939 ±0.008 | 0.931 ±0.014 |
| 1.0 | 91.12 ±0.23 | 0.895 ±0.036 | 0.901 ±0.029 |

Table 3: Accuracy and AUROC trend with λ for model trained on Fashion-MNIST.

| α | Accuracy | AUROC(M) |
|---|---|---|
| 0.01 | 91.89 ±0.03 | 0.928 ±0.002 |
| 0.1 | 92.23 ±0.08 | 0.951 ±0.004 |
| **0.2** | 92.36 ±0.13 | 0.953 ±0.004 |
| 0.5 | 91.15 ±0.19 | 0.918 ±0.005 |

Table 4: Accuracy and AUROC(M) trend with $\alpha$ in GP Loss on Fashion-MNIST

## 4.3 CIFAR-10

Till now, we have validated the central claim of the paper on the Fashion-MNIST dataset, in this subsection we provide few experiments to validate the claims made by authors about DUQ on CIFAR-10.

| Method | Accuracy | AUROC | Train time(in sec) | Inference time(in sec) |
|---|---|---|---|---|
| Deep Ensemble | 94.44 ±0.42 | 0.949 ±0.003 | 300 | 14 |
| DUQ ($\lambda = 0.5$) | 93.45 ±0.32 | 0.931 ±0.003 | 210 | 4 |

Result comparison on model trained on CIFAR-10 by deep ensemble and DUQ, AUROC reported is for separating CIFAR-10 from SVHN (averaged over 3 runs)

In Table 4.3, we compare DUQ with deep ensemble trained on CIFAR-10, the deep ensemble performs slightly better(1%) than DUQ in terms of accuracy and AUROC however DUQ is better in terms of computational time. We get similar values as in the original paper with a deviation of 0.5% in the values.

We obtain a similar rejection classification plot for CIFAR-10 vs SVHN as reported in the paper i.e. we see both DUQ and deep ensemble capture equal area as shown in Fig 9.

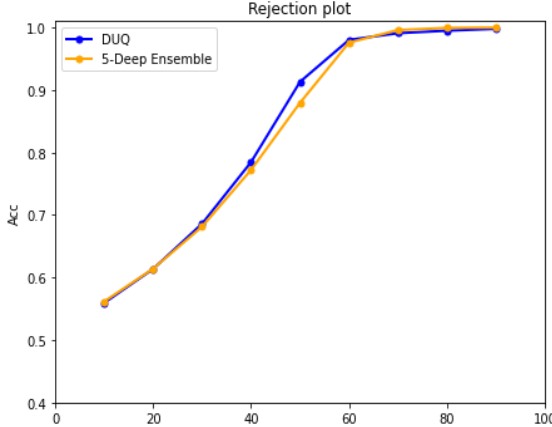

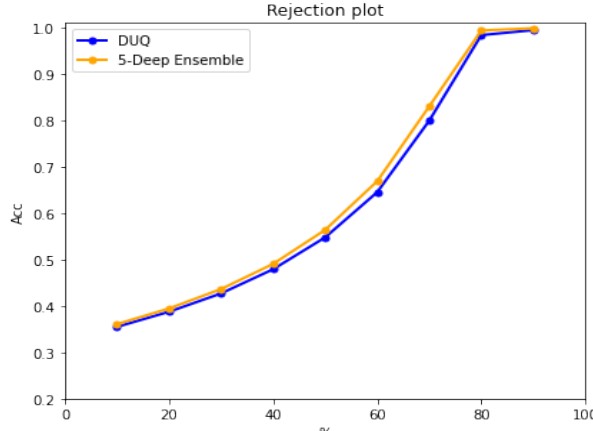

Figure 8: Rejection classification plot for deep ensemble and DUQ for Fashion-MNIST vs MNIST, x-axis is the % of data rejected using uncertainty estimates. Y-axis is the corresponding accuracy of prediction.

Figure 9: Rejection classification plot for deep ensemble and DUQ on CIFAR-10 vs SVHN. (X-axis is the % of data rejected using uncertainty estimates. Y-axis is the corresponding accuracy)

We provide a prediction histogram of DUQ on CIFAR-10 vs SVHN as shown in Fig 10. As datasets are of different sizes, we made the frequencies of histograms in proportion to dataset size as done by authors. We get the same graph as in the paper where the plot for CIFAR-10 is skewed towards the right extreme depicting higher certainty whereas a uniform plot for SVHN indicates uncertainty. These results support the claims on CIFAR-10 from the paper.

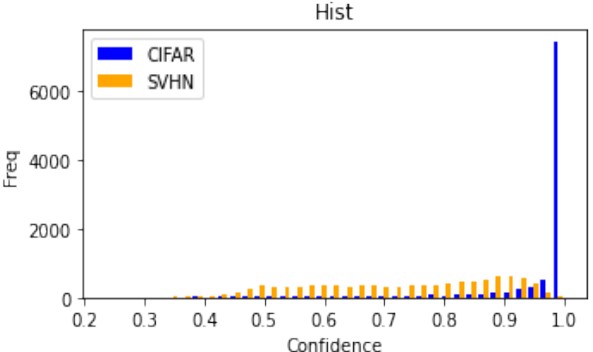

Figure 10: Uncertainty histogram of CIFAR-10 and SVHN for DUQ ($\lambda = 0.5$) trained on CIFAR-10

Additionally, we provide an analysis of the behavior of DUQ on noisy CIFAR-10 data points. In Figure 11, Noisy CIFAR-10 is CIFAR-10 added with a zero-centered gaussian noise with an std of 0.1. And all plots are made in proportion to the dataset size. We run the experiment with both the models and can see that DUQ is unconfident with Noisy CIFAR-10 points whereas deep ensemble can classify them with certainty, this finding is in line with that of Section 4.2.1.

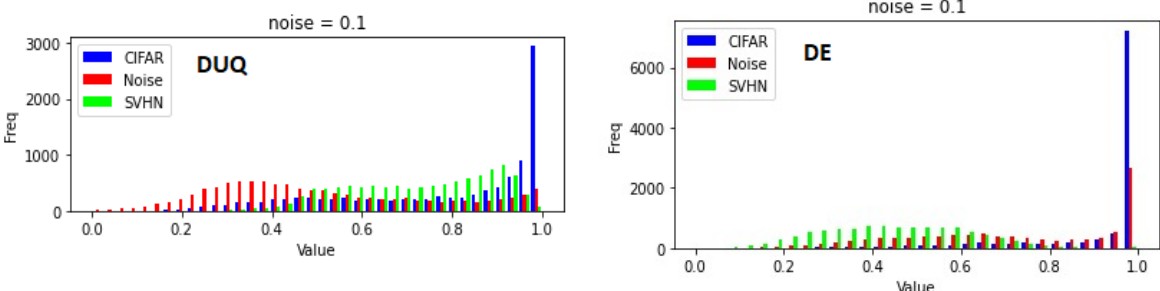

Figure 11: Uncertainty histogram of CIFAR-10 , SVHN and Noisy CIFAR-10 for DUQ ($\lambda = 0.5$) and deep-ensemble trained on CIFAR-10

# 5 Discussion

A simple approach along with data and models involved being in our computational limits helped us in the reproduction of results. We can validate most of the results and trends as reported in the paper.

Section 4.2 shows the overall behavior of DUQ and validates most of the claims made by authors like the importance of two-sided gradient penalty in the loss for enforcing the desired sensitivity which can show the state of the art OoD detection performance with a modest RBF network. Our additional experiments for understanding the nature of sensitivity of DUQ show how the addition of noise which is imperceptible to human eyes makes DUQ vulnerable for practical purposes.

Paper mentioned work on aleatoric and epistemic uncertainty detection is required, our naive extension of predicting $\sigma$ inspired by [3] can even outperform the author's results on Fashion-MNIST along with explicit control over uncertainty detection which is required for practical purposes. These preliminary results are encouraging for future research to be done in this direction.

Section 4.3 shows the scalability of different analyses for DUQ on larger datasets.

Overall we conclude that our reproduction results support different claims by authors of DUQ, it is an easy to implement, time-efficient algorithm with high sensitivity to input making it a good choice for OoD detection.

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
