# OpenReview forum: "Uncertainty Estimation Using a Single Deep Deterministic Neural Network - ML Reproducibility Challenge 2020"
_ML_Reproducibility_Challenge/2020 — Reject_

### Official Review · AnonReviewer3 · 2021-02-25
**Review of DUQ for ML Reproducibility Challenge 2020**

**Rating:** 7
**Confidence:** 3

**Review:**

# Reviewer Guidelines Rubric

---

* *Reproduciblity Summary:* Followed template
* *Scope of reproducibility*: Is stated and adhered to.
* *Code*:
  * Used original authors code
  * Additional experiment code is included as supplemental material
  * Code looks reasonable, `README.md`  and   `requirements.txt`  included.
* *Communication with Original Authors*: mentioned briefly, sounds cordial, only appears on summary.
*  *Hyperparameter Search*
  * Reused many parameters from appendix of original paper.
  * Did search over $\lambda$
  * Also experimented with centroid size and penalty constant $c$ as hyperparameters
  * Only tested one hyper parameter at a time instead of simultaneously
* *Ablation Study*:
  * I believe some parts of the hyperparameter search accomplish this (eg  $\lambda = 0$) but otherwise not discussed.
* *Discussion on Results*:
  * Reproduced results closely on several standard data sets
  * Reproduced the baseline model (Deep Ensemble) as well DUQ
  * No side-by-side comparisons with original paper
  * Other than on summary page, does not really describe which parts were easy or difficult to reproduce.
* *Recommendations for reproducibility*:
  * Not provided
* *Results beyond the paper*:
   * Introduced extension E-DUQ and some preliminary exploration
* *Overall organization and clarity*
  * Looks fine in general
  * LaTex is annoying sometimes, but would be good to get plots/tables on same page they are discussed somehow.

---

### Summary

This paper does accomplish the goal of reproducing the original paper to a large degree, but the discussion of reproducibility is short - it would be stronger if there were more in-depth explanation of what was easy, what was hard, and what the original authors could improve.

It is also difficult to judge how successful the reproducibility exercise is because the authors do not provide a side-by-side comparison. The text could quote the original results directly rather than alluding to them ambiguously.

This paper does reproduce the baseline model used for comparison, which strengthens its results and is appreciated.

The extension E-DUQ could be useful, but isn't explored enough in this paper and is only tested on one example. It might be better to expand it as a stand-alone paper instead.


**Familiar With The Original Paper:**

I have not read the original paper

**Reproducibility Summary:**

Report has summary

---

### Official Review · AnonReviewer2 · 2021-02-26

**Rating:** 5
**Confidence:** 3

**Review:**

__Summary__

The original paper (OP) proposes a deep learning  architecture based on a single deterministic neural network that produces uncertainty estimates. The OP proves the performance of the method in a toy setting (2 moons) and 2 usual OOD classification problems (MNIST and CIFAR10). The OP also studies slight modifications of the proposed loss function and the consequences on performance of their method.
The reproducibility report (RR) reproduces most of the experiments of the OP. Additionally, the RR performs an additional experiment to better illustrate one of the stated limitations of the paper (namely its inability to separate espistemic from aleatoric uncertainty).
Finally, the RR proposes an extension to the method from OP by learning a parameter previously fixed.

__Positive points__

- The RR reproduced most of the experiments from the OP and managed to achieve similar results. As a consequence, the RR's authors confirmed that they were able to generally reproduce the results from the OP and validate its claims.
- The RR's authors made the effort to contact the OP's authors to show them their report, which is very positive.
- The RR added a figure (Fig. 1) of the architecture which was not present in the OP. This is helpful in understanding the neural network studied and the mechanism proposed.
- The RR discussed clearly what kind of hardware was used and the time required to perform the computations, which is of great use to the reader trying to reproduce the results of the OP.
- The RR performed an additional experiment on both the MNIST and CIFAR10 datasets, showing that DUQ was not robust to a noise perturbation of the input. This experiment clearly and explicitly confirms a limitation mentioned by the OP.
- The RR proposes an extension to the original idea to try to alleviate the limitation mentioned above. The RR then performs experiments illustrating the performance of this extension and comparing it with DUQ. By doing so, the authors of the RR have shown their understanding of the paper. Proposing a new method is a great initiative.


__Negative points__

- The RR claims that all reproduced results are matched within 1% accuracy, but these quantities are never computed during the paper. Worse, this claim appears to be false (Tb. 5 of the RR vs Tb. 1 of the OP). This makes trusting the conclusions of the report difficult.
- A clear conclusion on what was being reproduced and the state of reproducibility is often missing. Sec 4.1: what are you trying to reproduce? Are you satisfied with the result? Sec 4.2: the RR mentions that the trend is similar: can you ground this statement with a quantified metric? Moreover, at l.122, the RR mentions that "[their] analysis supports the claims made by [OP] authors". It would be useful to precise which claims exactly are being validated by the RR analysis. Same at l.173. In addition, some statements are made regarding the reproducibility but are often vague (Sec 4.2, l.122: "slightly more accuracy per rejection", Sec 4.3: similar rejection classification plot, similar performance, etc).
- The RR could have been better organized. The order of the figures does not match the progression of the discussion (for instance, Fig.8 is reference directly after Fig.1). The same can be said for tables. Additionally, the RR could have benefited from better formatting (Sec 3.1 especially). This overall makes the RR difficult to read.
- It is sometimes unclear what is the contribution of the RR and what is the contribution of the OP. For instance, is Tb. 3 (Sec 4.2.3) trying to reproduce the OP results or is it an additional results? The same can be said for the experiment on c from Sec 4.2.3. In Sec. 4.2, you compute the computational time taken by the deep ensemble and DUQ. Is it your own experiment? Is it a reproduction from the OP? If then, what is your conclusion on the reproducibility?
- Limited details on the training procedure for the additional experiments is given. A description or at least a pointer to the RR code would have been appreciated. Moreover, background on some choices (for instance, why the value of sigmas in Fig.6 and Fig.7?) would have helped understand the design choices.


===

Overall, I find that this was an encouraging report that addressed the general reproducibility of the paper. I appreciated the initiative to propose new experiments.
Reorganizing the report would greatly improve its readability. While the new experiments were interesting, I found that the report would benefit from a deeper discussion on the current state of reproducibility. Adding details on the methodology of the experiment, what the RR authors set to reproduce in this specific experiment, and a clear conclusion on the reproducibility of the experiment would also improve this report to help it make a more grounded statement on the status of reproducibility of the paper in general.

===

__Additional comments__

- Fig. 1 is very helpful, but takes too much space in the RR. Consider making it smaller.
- Is it not clear what is i in the equations of Sec 3.1. The summation signs do not indicate what variable is being summed.
- In Tb. 1: is the training time a new result? A reproduced one? This is not said clearly.
- The authors study changing the constant number in the two-sided gradient penalty (Sec 4.2.3). At the first reading, it is not clear what exactly is changed since the letter c is already used in the precise equation of the gradient, but not with the same meaning as in Sec 4.2.3. Additional precision on the exact quantity that is being changed and additional care to the notation could improve the understanding of what the RR tried to do.
- The term "sensitivity" in Sec 3.1 is not defined.


**Familiar With The Original Paper:**

I have not read the original paper

**Reproducibility Summary:**

Report has summary

---

### Official Review · AnonReviewer1 · 2021-03-10
**An OK report, but confidence intervals for tested metrics is a must (otherwise hard to make reliable comparisons and ablations)**

**Rating:** 6
**Confidence:** 3

**Review:**

The authors test two claims made by the original paper: the ability of the proposed approach to detect OOD, and the role of different hyperparameters as an ablation study. In addition to that, the authors explore sensitivity of the proposed algorithm to noise, and propose an extension for explicit detection of aleatoric and epistemic uncertainty.

* Re noise sentivity of the algorithm (Figure 6), it is quite surprising that Deep Ensemble or E-DUQ perform so much worse that DUQ, even when we reject 95% of data. Can the authors give more details and better explain why that is the case?
* In Table 1-5, how many splits (or randomly generated train datasets) were used?
* Please provide confidence intervals in tables (e.g. 5-95% quantiles) for performance metrics. Otherwise, it is hard to assess what is significant. For AUROC, you need to report bootstrapped values.
* More details about lengthscale prediction would be needed (section 4.2.2)
Minor:
* The authors make several grammatical mistakes, e.g., in the usage of articles "the". The abstract has no subject in their sentences, periods are often separated from the final word. Overall, although the text is understandable, the writting would need some polishment, and it would be nice if the English could be checked by a spell checker.
* Eq. 54-55, what is m_{c,t}? Not defined in the text.
* The explanation about the gradient penalties is not self-contained, that is, it is not understandable without having to go back to the original paper, so either remove it or make it self-contained.
* Because this is a reproducibility challenge, the authors should provide more details on experimental setup to generate Tables 1-5 (what hyperparameters, setups, splittings, number of runs, etc, etc).

**Familiar With The Original Paper:**

I have not read the original paper

**Reproducibility Summary:**

Report has summary

---

### Decision · Program_Chairs · 2021-03-31

**Decision:**

Reject

**Comment:**

Overall reviews and/or the paper content not good enough for the AC to recommend to the journal.